# The Party Wall: Redefining the Indications of Transcranial Approaches for Giant Pituitary Adenomas in Endoscopic Era

**DOI:** 10.3390/cancers15082235

**Published:** 2023-04-10

**Authors:** Sabino Luzzi, Alice Giotta Lucifero, Jessica Rabski, Paulo A. S. Kadri, Ossama Al-Mefty

**Affiliations:** 1Department of Clinical-Surgical, Diagnostic and Pediatric Sciences, University of Pavia, 27100 Pavia, Italy; alice.giottalucifero01@universitadipavia.it; 2Neurosurgery Unit, Department of Surgical Sciences, Fondazione IRCCS Policlinico San Matteo, 27100 Pavia, Italy; 3Department of Brain and Behavioral Sciences, University of Pavia, 27100 Pavia, Italy; 4Brigham and Women’s Hospital, Harvard Medical School, Boston, MA 02115, USA; 5Medical School, Federal University of Mato Grosso do Sul, Campo Grande 79070-900, Brazil

**Keywords:** cavernous sinus, cranio-orbito-zygomatic approach, endonasal endoscopic approach, parasellar tumors, pituitary adenoma, pterional approach, trans-sphenoidal approach

## Abstract

**Simple Summary:**

Do transcranial approaches still play a role for giant pituitary adenomas, given the marked evolution of endoscopic endonasal trans-sphenoidal skull base surgery? This narrative paper reviews the key arguments through a critical appraisal of the personal series of the senior author (O.A.-M.). Traditional indications for transcranial approaches include the absent pneumatization of the sphenoid sinus; kissing/ectatic internal carotid arteries; reduced dimensions of the sella; lateral invasion of the cavernous sinus lateral to the carotid artery; dumbbell-shaped tumors caused by severe diaphragm constriction; fibrous/calcified tumor consistency; wide supra-, para-, and retrosellar extension; arterial encasement; brain invasion; coexisting cerebral aneurysms; and separate coexisting pathologies of the sphenoid sinus, especially infections. Residual/recurrent tumors and postoperative pituitary apoplexy after trans-sphenoidal surgery require individualized considerations. Transcranial approaches still play a critical role in giant and complex pituitary adenomas with wide intracranial extension, brain parenchymal involvement, and the encasement of neurovascular structures.

**Abstract:**

The evolution of endoscopic trans-sphenoidal surgery raises the question of the role of transcranial surgery for pituitary tumors, particularly with the effectiveness of adjunct irradiation. This narrative review aims to redefine the current indications for the transcranial approaches for giant pituitary adenomas in the endoscopic era. A critical appraisal of the personal series of the senior author (O.A.-M.) was performed to characterize the patient factors and the tumor’s pathological anatomy features that endorse a cranial approach. Traditional indications for transcranial approaches include the absent pneumatization of the sphenoid sinus; kissing/ectatic internal carotid arteries; reduced dimensions of the sella; lateral invasion of the cavernous sinus lateral to the carotid artery; dumbbell-shaped tumors caused by severe diaphragm constriction; fibrous/calcified tumor consistency; wide supra-, para-, and retrosellar extension; arterial encasement; brain invasion; coexisting cerebral aneurysms; and separate coexisting pathologies of the sphenoid sinus, especially infections. Residual/recurrent tumors and postoperative pituitary apoplexy after trans-sphenoidal surgery require individualized considerations. Transcranial approaches still have a critical role in giant and complex pituitary adenomas with wide intracranial extension, brain parenchymal involvement, and the encasement of neurovascular structures.

## 1. Introduction

### 1.1. Historical Evolution of Pituitary Surgery

The historical evolution of pituitary surgery is fascinating. The name Galen of Pergamon is derived from "pituita" from the Greek “ptuo” (to spit) and the Latin “pituita” (mucus). Andreas Vesalius modified it to mean “glans in quam pituita destillat”, describing it as a small anatomical structure at the base of the brain at that time believed to secret products from the brain into the nasal cavities [1]. Based on the seminal observations taken by others and the early reported successful surgeries on the thyroid and parathyroid gland [2,3,4], Cushing, in 1909, highlighted the pituitary gland as a source of endocrinological illness and therefore a fundamental surgical target [5]. The first report of a pituitary adenoma might be of Andrea Verga in 1864 [6]. His autoptic observations described an abnormal pituitary growth compressing the optic nerves. After that, similar remarks focused primarily on the pathological substrate of acromegaly [7,8,9,10,11,12]. The birth of pituitary surgery in the early 1900 was facilitated by significant progress in endocrinology and radiology [13]. Arthur Chloffer used radiography to confirm the sellar pathology before performing the first reported successful trans-sphenoidal procedure [14]. This initial enthusiasm for trans-sphenoidal was tempered by Cushing, who reverted to a subfrontal approach to overcome the limits he noted at that time regarding a lack of adequate illumination and maneuverability with the trans-sphenoidal corridor. Hemostasis also appeared to be a problem. The trans-sphenoidal approach remained in use by notable surgeons Dr. Dott and Guiot [15,16,17,18,19]. However, it was reborn and gained popularity owing to the advancements introduced by Jules Hardy who used an operative microscope and radiological guidance [20,21,22,23,24,25,26]. Preoperative polytomography and CT scans permitted the anatomy, and its variations in the sella and sphenoid sinus, to be delineated [27,28] and facilitated navigation during surgery [29,30].

The recent explosion of endonasal endoscopic surgery and neuronavigation has made the trans-sphenoidal route overwhelmingly popular. At the same time, it is deficient in some instances, requiring a transcranial approach.

This article attempts to define the transcranial approaches that can be used for pituitary tumors.

### 1.2. Arguments for Surgery 

Pituitary adenomas have a prevalence ranging between 15% and 17% in the general population and are the third most frequent type of intracranial tumor in adults [31,32,33]. They are treated for mass effects or endocrinological disorders. While prolactinomas respond to medical therapy, microadenomas secreting GH or ACTH can be cured nowadays with endoscopic endonasal trans-sphenoidal surgery. 

Pituitary adenoma surgery has three goals: the decompression of optic pathways, the preservation or restoration of the pituitary function, and the prevention of malignant transformation. The latter is known to significantly, but not exclusively, affect syndromic and sporadic ACTH-secreting tumors and prolactinomas, and can be characterized by a poor prognosis [34,35,36,37,38]. The symptomatic progression and recurrence rate proved to be lower in the case of gross total versus subtotal resection [39,40,41,42]. Redo surgery is recommended for recurrent or residual nonfunctioning pituitary adenomas [43]. While recognizing the current role of radiotherapy for the long-term control of the tumor residual, the pursuit of radical resection, whenever possible, is justified by eliminating the risk of radiation or minimizing the residual for safer radiosurgery, particularly concerning the optic nerve and chiasm [44,45].

### 1.3. The Party Wall

The sellar region is a party wall between the neurocranium and splanchnocranium. Giant pituitary adenomas represent a complex clinical entity whose surgical management often imposes the choice between a trans-sphenoidal versus transcranial approach. The evolution of endoscopic transnasal trans-sphenoidal surgery and its good results in pituitary tumors have overshadowed the need for transcranial approaches in particular circumstances, mainly for giant pituitary adenomas. 

This narrative review aims to define the contemporary indications for transcranial approaches in giant pituitary adenomas by analyzing specific anatomical factors related to the patient and anatomopathological aspects associated with the tumor. In addition, technical innovations associated with future directions of the surgical management of giant pituitary adenomas are also discussed.

## 2. Factors Related to The Inter-individual Anatomical Variability of the Sellar and Parasella Area 

### 2.1. Pneumatization of The Sphenoid Sinus

In 1920, Congdon first highlighted the surgical relevance of the *“distribution and mode of origin of septa and walls of the sphenoid sinus”* [46]. The pneumatization process starts during the third year of life and is highly variable [47,48]. It proceeds in a ventral-to-dorsal direction from the anterior part of the sphenoid body to the occipito-sphenoid suture [49]. Different classifications have been reported that aim to anticipate the complexity of the exposure of the sellar floor during the trans-sphenoidal approach [50,51]. Hamberger et al.’s original paper reported a frequency of the conchal type of 3% in terms of sagittal pneumatization [52]. Similar data have been reported by others [49,50,51,53,54]. Recently, Lazaridis and associates reported a frequency of 4% [55] in a more detailed cadaveric study. There is also a rationale to believe that this value remains underrated because of the different measurement techniques used in human studies. The merit of the more recent classifications of the sphenoid sinus was mainly to focus on the degree of coronal pneumatization, which is essential during the planning of the trans-sphenoidal approach [50,51,53].

Conchal-type sinuses and other mixed patterns of the coronal non-pneumatized sphenoid sinus are generally believed to be less favorable for a trans-sphenoidal approach, especially in children and large tumors. Flat sellar-type and complex sphenoid sinus configurations, with incidences of 11% and 29%, respectively [56], are also considered challenging for the trans-sphenoidal route. Although these anatomical variants present surgical difficulties, they do not act as counterindications for endonasal endoscopic surgery or mandate transcranial surgery.

### 2.2. Angiomorphology of The Internal Carotid Arteries

"Kissing" internal carotid arteries (ICAs) may touch each other within the sphenoid sinus, sphenoid bone, or sella [57,58,59]. The intercarotid distance is a recognized limiting factor for the trans-sphenoidal approach [60]. Nevertheless, no data on the minimal safe distance have ever been reported, meaning that the trans-sphenoidal route is contraindicated. The average length reported in the literature ranges between 12 and 14 mm [61,62,63,64]. A distance of less than 10 mm is technically challenging [63]. Yilmazlar and colleagues found that coronal distances between both ICAs on cadaveric specimens were 17.1 ± 4.0 mm anteriorly, 20.3 ± 4.2 mm medially, and 18.8 ± 4.6 mm posteriorly. These measurements were 15.4 ± 1.8 mm, 16.0 ± 2.8 mm, and 16.2 ± 3.4 mm for anterior, medial, and posterior measurements, respectively, on normal sella MRI images [65]. They also found the distance between the cavernous carotid and the pituitary gland to be lower than 0.5 mm on 27% and 32% of the right and left sides, respectively [65]. Based on these data, the anterior segment, namely the paraclinoid ICA, is more susceptible to vascular injury during trans-sphenoidal surgery for microadenomas. The widening grade of the working corridor expected in giant adenomas is significantly smaller in non-giant ones, and this distance is even lower in females [65]. Apart from the intercarotid distance, the angiomorphology of the ICAs is pivotal since they can be tortuous and ecstatic, and can project toward the midline up to 4 mm to the contralateral one in 10% of cases, as reported by Renn and Rhoton [63]. These findings have been highlighted to be more frequent in acromegalic patients, severe hypertension, and polycystic kidney disease. A tragic amount of these were mistaken for tumor mass and were biopsied. Imaging evidence of dolichoectatic, tortuous, kissing cavernous, paraclival, or paraclinoid carotid arteries is a crucial factor in favor of the transcranial approach, as also stressed by other groups [66].

### 2.3. Dimensions of The Sella

Ten years before the advent of the CT scan, Sir Godfrey Hounsfield, Di Chiro, and Nelson reported the measurement of the sella turcica based on conventional radiography, noting that *“considerable enlargement of the pituitary may be unobservable roentgenologically”* [67]. This means that pituitary tumors are not always associated with an increase in the sellar volume. The same aspect was also examined by James Provenzale in 2006 in his editorial on the occasion of the first century in the *American Journal of Roentgenology* [68]. The size of the sella turcica correlates neither with the cephalic index, cranial diameters, nor the size of the sphenoid sinus [69]. The female gender has a lower volume of the sella [70]. A small sellar size has been postulated to play a role in the etiopathogenesis of Sheehan’s syndrome [71] and the volume of the sella turcica has been proven to decrease by 32% on average in those patients treated with long-acting somatostatin analog [72]. Critical dimensional data on the sella turcica involve its depth, length, width, and volume. The first three are still measured today based on the classic method of Taveras and Wood [73]. In contrast, the volume is derived by the simplified mathematical formula reported by Di Chiro and Nelson that approximates the sella to an ellipsoid: volume (cm^3^) = 0.5 (length × width × depth in mm)/1000 [67] (Figure 1). 

Renn, Rhoton, Di Chiro, and Nelson reported average sella turcica volumes of 621 mm^3^ and 594 mm^3^, respectively, with a broad standard deviation [63,67]. 

Because of limited surgical maneuverability, we believe that a sellar volume at the lower limits of the Gaussian distribution should be considered a disadvantage to a trans-sphenoidal approach. This point potentially informs the choice of a transcranial route, especially in large-to-giant adenomas with a significant extra-sellar extension. A non-negligible level of complexity for the trans-sphenoidal approach in these rare cases was also highlighted by Ouaknine and Hardy [74]. 

### 2.4. Vascular Pattern of Intercavernous Sinuses

A preoperative evaluation of the vascular patterns and sizes of the intercavernous sinuses is considered in order to choose between a trans-sphenoidal and a transcranial approach. Significant venous bleeding from the anterior and inferior intercavernous sinuses during the dural opening has been reported to cause limited exposure and incomplete tumor resection [75,76,77]. Winslow [78] first and Knott [79] later already found a complete circular sinus within the sella in all their corrosion cast specimens, the so-called “inferior circular sinus of Winslow”. Accordingly, Kaplan highlighted that "*the neurosurgeon should be aware of the possible presence of sizable intercavernous venous connections within the dural lining of the hypophyseal fossa*" [77]. Later, accurate anatomic–radiological correlative studies were reported by others [63,77,80,81,82,83].

Gadolinium contrast-enhanced magnetic resonance venography of the intercavernous sinuses is considered reliable in the preoperative planning of the trans-sphenoidal approach, with sensitivity rates of 37% and 48%, respectively, for the detection of the anterior and inferior sinuses [84]. Recently, Deng and co-workers classified the angioarchitectural patterns of the intercavernous sinuses into four types based on their combined autoptic and MRI findings [84]. The anterior and inferior intercavernous sinus frequencies were 78% and 61%, respectively, in the cadaveric cohort [84]. The anterior intercavernous sinus is seen on the axial plane, while the inferior intercavernous sinus is found on the coronal plane. Deng type 3 pattern, characterized by an anterior and inferior intercavernous sinus found in 50% of cadavers, was associated with a highly narrowed operative corridor in relation to the gland [84] (Figure 2).

Accordingly, despite effective hemostatic agents that control venous bleeding, encountering a large inferior intercavernous sinus can be difficult in a trans-sphenoidal approach, while this would not be encountered with a transcranial approach. 

### 2.5. Anatomy of The Sellar Diaphragm 

Large and giant pituitary adenomas can be resected with a single endoscopic procedure in less than 50% of cases [85,86,87,88,89]. The sellar diaphragm’s anatomy affects the suprasellar growth pattern of large and giant pituitary adenomas, and the invasion of the subarachnoid space is among the most important limiting factors of the gross total resection [86,89,90,91,92,93]. The diaphragm is defined as competent or incompetent based on the size of the opening, for which Rhoton’s group identified three different morphological patterns: type A (<4 mm), B (4–8 mm), and C (>8 mm) [94]. In the case of the incompetent diaphragm, as in type C, the tumor invades the subarachnoid space early, the encasement rate of the neurovascular structure is higher, and the chance for a gross total resection is lower with a trans-sphenoidal approach [93]. 

Figure 3 reports a case from the personal series of the senior author (O.A.-M.).

The huge suprasellar component of an invasive Hardy type C giant non-functioning pituitary adenoma invaded the subarachnoid space through a Rhoton type C incompetent diaphragm. The tumor involved the lamina terminalis, the subcallosal area, the foramina of Monro, and the third ventricle floor, causing initial hydrocephalus. The patient was a 40-year-old male suffering from chronic headaches and bitemporal hemianopsia. This eventuality is a classic indication of a transcranial approach.

When the diaphragm is totally or largely competent, as in types A and B, it can be displaced upward, confining the tumor in the infra-diaphragmatic space or trespassed by the tumor, which will show a supra-diaphragmatic component. The latter growth pattern constitutes the typical dumbbell shape (waist sign on MRI). Purely infra-diaphragmatic tumors are easily managed with a trans-sphenoidal corridor. Conversely, adenomas with a large and giant dumbbell shape impose a trans-tubercular extension of the trans-sphenoidal approach. It should involve trans-diaphragmatic access to reach the suprasellar area since, regardless of the consistency, the likelihood of a descent supradiaphragmatic component has been reported to be considerably lower in the case of significant diaphragmatic constriction [86,93,95]. A neck-to-dome cutoff ratio of 1.9 has been proven to be a reliable prognostic factor of non-descent tumors with a sensitivity of 77% [95]. The smaller the neck-to-dome ratio, the greater the diaphragmatic constriction. Nondescent remnants involve three consequences: first, the extent of resection (EOR) is significantly decreased [91]; second, the risk of postoperative bleeding is higher [95]; and third, the risk of a cerebrospinal fluid leak is increased. Accordingly, a transcranial approach should be preferred for dumbbell-shaped large and giant pituitary adenomas. This indication is more significant as more of a marked diaphragmatic constriction. 

Figure 4 shows the persistence of wide suprasellar remnants in a patient who harbored a giant Hardy type C pituitary adenoma that underwent a trans-sphenoidal approach. In this case, the diaphragm’s anatomical type was coherent with that of Rhoton’s type A (<4 mm), and the neck-to-dome ratio was 0.3.

#### Case #1: Infradiaphragmantic Tumor with Middle Fossa Involvement through the Cavernous Sinus

A 39-year-old female suffering from headache, right-sided facial pain and numbness, and hyposmia was diagnosed with a non-functioning giant pituitary adenoma extending bilaterally into the cavernous sinus (CS). The mass also significantly thinned the dorsum sellae and eroded bilateral petrous apices (Figure 5). 

Because of a competent diaphragm (Rhoton type A), the tumor was mostly infra-diaphragmatic. However, a non-negligible mass component involved the right middle fossa through the roof of the CS. A staged resection of the adenoma was planned, where an extended trans-sphenoidal corridor with right medial maxillotomy was used to remove the sellar component of the lesion (Figure 6).

Six months later, the second stage was performed after the a new MRI scan which showed a volumetric increase in the known intracranial residue (Figure 7). 

The patient underwent gross total tumor resection through a cranio-orbito-zygomatic (COZ) approach (Figure 8).

## 3. Factors Related to The Tumor Features

### 3.1. Tumor Texture and Consistency

The consistency of a tumor is a crucial factor impacting the EOR, complication rate, and outcome of giant pituitary adenomas [89,90,91,96,97,98,99]. Higher-consistency tumors tend to be non-descending when supradiaphragmatic and associated with an incompetent diaphragm. Often, they are non-descending even when infra-diaphragmatic. Higher-consistency tumors are unaffected by facilitating maneuvers, such as jugular vein compression, Valsalva, or air or saline injection through the lumbar drain. Yamamoto and colleagues reported that up to 15% of giant pituitary adenomas might be firm, fibrous, or calcified, necessitating an expanded trans-sphenoidal or transcranial approach [100]. Besides the surgical planning and approach selection, the anticipation of the tumor’s consistency also affects the resection technique, which may take advantage of using an ultrasonic aspirator or especially arachnoidal dissection. In the series of Rutkowski et al., the intraoperative frequencies of the “partially suckable” (grade 3) and “not suckable” (grade 4) adenomas were 40.8% and 17%, respectively [100]. Grades 4 and 5 cause more frequent hypopituitarism [101], regularly invade the CS, have a higher rate of intraoperative cerebrospinal fluid leaks, are characterized by a less significant improvement in preoperative visual dysfunction, and show a lower gross total resection rate [100]. As a result of these aspects, the efforts of radiology have been directed at identifying those imaging features that are critical in the preoperative differential diagnosis between “soft” and “firm” adenomas. The discriminative parameter has typically been the different histologic content of the collagen [102,103,104,105,106]. The relative signal intensity or signal intensity ratios on T1- or T2-weighted MRI and diffusion-weighted apparent diffusion coefficient (ADC) values have been widely studied, demonstrating, however, conflicting results [100,101,102,107,108,109,110]. Over the last few years, dynamic contrast-enhanced MRI (DCE-MRI) with pharmacokinetic analysis has attracted attention to quantitatively assess collagen content [111,112,113,114]. Kamimura and co-workers found that firm tumors have a greater volume of extravascular extracellular space per unit volume of tissue (ve) on DCE-MRI because of a higher percentage of collagen content [114]. This dynamic MRI parameter has been proven to have a very high predictive value for the consistency of pituitary adenomas. 

Other promising techniques include MRI elastography, and high-resolution ADC achieved with 7 tesla diffusion-weighted MRI imaging [109,115].

### 3.2. Suprasellar Extension

Based on his roentgenographic studies, in 1940, Geoffrey Jefferson defined the extrasellar growth of giant pituitary adenomas as *“unpredictable”*. The development pattern is affected by several factors, such as tumor progression, the state of fixation of the chiasm, the pituitary fossa’s shape, and the sellar diaphragm’s specific anatomy [116]. Jules Hardy, in the 1970s, included the possible symmetrical (type 0–C) or asymmetrical (type D) suprasellar extension of pituitary adenomas in his classification of invasiveness [117,118,119]. Giant pituitary adenomas can expand significantly in the suprasellar region when their growth pattern mainly involves the midline (cases #2–#8). On the other hand, the tumor can develop in one or various directions simultaneously, eroding the planum sphenoidale or the clivus in case of subfrontal or retrosellar extension, respectively (sellar invasion grade IV [118]). Hardy grade C or D suprasellar extension has been reported to strongly predict a reduced gross total resection (OR 3.91, P = 0.01) [120]. The volume per se of the suprasellar portion of the tumor has been proven not to affect the EOR rate of the trans-sphenoidal corridor [89,90,96,98]. Nevertheless, a clear distinction should be made between tumors that displace the optic chiasm and the complex of the anterior cerebral artery through a more or less competent diaphragm and those that instead encase the neurovascular structures and enroach the hypothalamus, the foramen of Monro, the third ventricle and lateral ventricles, and the columns of the fornix. For the latter, unavoidably causing hydrocephalus, a pterional approach or combined pterional/interhemispheric-transcallosal approach is recommended since worse outcomes have been reported with the trans-sphenoidal route [92,121,122]. 

Type D tumors involving the anterior cranial fossa deserve further consideration regarding the possible encasement of the optic chiasm and optic nerves. It is without a doubt that trans-sphenoidal surgery allows for an effective decompression of the optic apparatus, which is responsible for improving visual function in visually symptomatic patients. Nevertheless, a wider working corridor and a greater microscopic magnification are required in those cases where a circumferential encasement of the optic nerves is ascertained preoperatively to avoid surgical dissection, which may be a potential source of iatrogenic damage if the arachnoid plane is absent. The encasement of optic apparatus is a recognized potential source of visual loss after trans-sphenoidal surgery for dumbbell-shaped suprasellar tumors [123]. Tumors growing eccentrically under the frontal lobe also require an open approach [66,124]. 

#### 3.2.1. Case #2: Involvement of the Anterior Skull Base

A 43-year-old male patient suffering from headache, progressive visual loss in the left eye, decreased libido, fatigability, and an intolerance of cold was diagnosed with a giant Hardy type D non-functioning pituitary adenoma involving the subcallosal area and the anterior cranial fossa. The lesion encased the optic apparatus and pushed the anterior cerebral artery complex upward. On the FLAIR MRI, the tumor also showed a partial parenchymal invasion of the basal forebrain (Figure 9). 

A gross total resection of the tumor was performed through a right COZ approach, where a meticulous microneurosurgical dissection of the encased optic nerves and chiasm was performed. A small remnant invading the ventrostriatum was left, followed by neuroimaging (Figure 10). 

No recurrences occurred after 24 years and the remnant remained stable. 

#### 3.2.2. Case #3: Paramedian Pattern of Growth with Parenchymal Invasion

A 28-year-old male had a giant Hardy type D growth hormone and a giant pituitary adenoma involving the left CS and a substantial suprasellar, retrosellar, and left parasellar middle fossa extension. The lesion invaded the left cerebellar peduncle and showed cystic degeneration at its apical portion, compressing the ipsilateral temporal horn of the lateral ventricle (Figure 11). 

A right COZ approach was executed and the tumor was resected (Figure 12). 

Ten days later, the patient was discharged without deficits (Figure 13).

### 3.3. Parasellar Extension 

The medial wall of the CS is formed by a thin sellar part and a thicker sphenoidal part that, contrary to the lateral wall, are not easily separable. The sphenoidal part is frequently interrupted so that the intracavernous carotid is in direct contact with the pituitary gland in 52.5% of cases [125,126]. Consequently, giant pituitary adenomas very frequently show an involvement of the CS at diagnosis (cases #1, #3–#8), the amount of which is known to negatively affect the rate of gross total resection of the trans-sphenoidal corridor [127,128]. An invasion of the CS should be thoroughly considered in Knosp 3A, 3B, and 4, as proven by the intraoperative endoscopic inspection [129,130]. Moreover, 3T MRI imaging is more sensitive and specific than standard 1.5T MRI for the medial wall of the CS [131]. Proton-density-weighted imaging (PDWI), VIBE, contrast-enhanced sampling perfection with application-optimized contrasts using different flip-angle evolutions (CE SPACE), and fast spin echo (FSE) sequences have been proposed for the recognition of wall defects through which the tumor can invade the parasellar compartment [132,133,134,135]. Recently, radiomics-based methods have been proposed to assess the invasion rate on contrast-enhanced T1 and T2-weighted MRI [136,137]. However, the progressive implementation of angled endoscopes and extended transpterygoid and transmaxillary approaches have widened the reach of the trans-sphenoidal route, allowing access to the infratemporal fossa and lateral retroclival regions [138,139]. An extension of the adenoma beyond the CS and parasellar area remains one of the most evident indications of the transcranial approaches. An encasement of the ICA seen in all Knosp 4 adenomas makes the endoscopic endonasal approach unsafe and the gross total resection of the tumor unfeasible [121]. Recently, Castle-Kirszbaum and coworkers clarified that Knosp grade 3 or 4 CS invasion is a predictor of a reduced gross total resection (OR 7.62, P < 0.01), similar to Hardy grade C or D suprasellar extension [120]. Accordingly, their proposed Hardy, Age, Clival, Knosp, and Depth (HACKD) score for predicting the EOR of pituitary macroadenomas through an endoscopic endonasal trans-sphenoidal approach was also based on these critical points [120]. The extended trans-sphenoidal transpterygoid approach provides limited exposure of the portion of the tumor lying lateral to the ICA [140]. The pterional or COZ approach, based on the surgical maneuverability required to safely remove the parasellar part of the tumor [141,142,143,144,145], has been used as a standalone or staged combination with the trans-sphenoidal corridor. 

Usually, the remnant in the CS is treated with medical therapy or radiosurgery [146,147,148]. Not infrequently, giant parasellar tumors extradurally or intradurally involve the paraclinoid or supraclinoid ICA, which is another reason to use a transcranial approach. The total or partial encasement of the ICA, as well as ophthalmic, posterior communicating, and anterior choroidal arteries, requires a careful microsurgical dissection and suitable surgical maneuverability, easily achieved through a transcranial corridor. Guinto and Kitano also recommended the use of a transcranial corridor in cases where the cavernous component of the tumor is more significant in size than that which can safely undergo radiosurgery, usually <3 cm [140,149]. The same group also reported a better functional outcome of the oculomotor nerve in the case of encasement around the anterior clinoid, causing a preoperative deficit [149]. Adenomas growing eccentrically on the coronal plane under the temporal lobe should be treated with an open craniotomy, as stressed by Goel [124]. In the senior author’s practice and experience, the removal of a pituitary adenoma invading the CS is highly achievable, and, compared with other tumors, pituitary tumors are an easier surgical endeavor, with a smaller risk of both carotid injury and postoperative cranial nerve deficits, and a higher likelihood of being able to recover from a pre-existing cranial nerve dysfunction [150].

#### 3.3.1. Case #4: Involvement of the Middle Skull Base with Parenchymal Invasion

A 51-year-old female underwent a brain MRI for visual loss (R>L) that worsened over the last 2–3 months. MRI revealed a giant pituitary adenoma with a significant parasellar, suprasellar, retrosellar, and parenchymal extension (Figure 14 and Figure 15). 

Not surprisingly, by considering the laterality of the lesion, the patient had lateral homonym hemianopsia (Figure 16). 

A staged resection was planned to entail a sublabial trans-sphenoidal resection of the intrasellar part and a left COZ approach aimed at a gross total resection (Figure 17). 

Postoperatively, the patients had a partial deficit of the left third and fourth cranial nerves that improved after one month (Figure 18). 

#### 3.3.2. Case #5: Extension beyond the Lateral Wall of the Cavernous Sinus

A 58-year-old female was affected by a giant Hardy type D pituitary adenoma with an asymmetrical growth pattern. The patient suffered from severe nausea, vomiting, left hemiparesis, and right third nerve palsy. The mass invaded the CS that was significantly extended beyond its lateral wall in the middle fossa. The cavernous carotid and right P1 segment of the posterior cerebral artery were encased, and the lesion also involved the right tentorial incisura and the petroclival region (Figure 19).

A transcranial approach was indicated in this case and a COZ approach was performed.

#### 3.3.3. Case #6: Encasement of the Posterior Communicating and Anterior Choroidal Artery

A 52-year-old female asymptomatic patient was incidentally diagnosed with a giant pituitary adenoma. The tumor showed a significant extra-sellar component in the suprasellar and sphenoid region and the left CS. The posterior communicating and anterior choroidal arteries were encased due to the involvement of the supraclinoid segment of the ICA (Figure 20). 

A left pterional approach was performed, achieving a near-total (>90%) resection of the tumor (Figure 21). 

The patient had a partial third cranial nerve deficit that recovered after three months (Figure 22). Pathology was coherent with TSH-secreting adenoma. 

### 3.4. Retrosellar Extension

Not infrequently, sellar invasion Hardy grade IV and suprasellar extension Hardy grade C or D adenomas may involve the retrosellar or retroclival area. Grade IV tumors are characterized by extensive sphenoid bone erosion and dura erosion. This aspect is considered a feature of aggressive behavior [151] (cases #2–#5, #7, #8). 

Figure 23 shows another case from the personal series where the sphenoid bone and clivus invasion carried out by a giant Hardy type D pituitary adenoma is visible in a 21-year-old male acromegalic patient. 

Scheithauer and colleagues reported that the estimated rate of gross invasion of the neighboring structures by pituitary adenomas of all types is approximately 35% [152]. These “invasive” or “aggressive” tumors are more prone to encase the posterior circulation vessels, including the perforating arteries from the basilar artery and the thalamoperforating arteries from the posterior cerebral arteries.

Despite the tremendous advancements in endoscopic techniques and the development of sophisticated corridors able to reach the retrosellar or retroclival space [153,154], evidence on the encasement of posterior circulation vessels and cranial nerves remains a strong indicator for choosing a transcranial petrosal approach [155,156,157] that can achieve a safe gross total resection, especially in recurrent or previously irradiated tumors because of the adhesions or scar tissue [123].

#### Case #7: Encasement of the Paraclinoid and Supraclinoid Internal Carotid Artery

A 74-year-old female experienced a progressive decrease in visual acuity after cataract surgery and proptosis. MRI showed a giant pituitary adenoma with a left-side involvement within the anterior cranial fossa and ethmoid bone, as well as a wide retrosellar, clival, and retroclival extension. The mass also invaded the left CS, encircling the cavernous and petrous segment of the ICA (Figure 24).

Pathology function was normal. A staged trans-sphenoidal–transcranial approach was planned. The microscopic sublabial trans-sphenoidal step allowed the tumor’s sellar part and a large part of the clival component to be resected. The retrosellar portion was approached instead with a pterional corridor through a left subfrontal perspective. 

This route allowed a precise and safe arachnoidal tumor dissection at the interpeduncular and prepontine cisterns (Figure 25).

Pathology was consistent with an ACTH-secreting pituitary adenoma. 

### 3.5. Arterial Encasement

The evidence of arterial encasement can alter the overall management of a giant pituitary adenoma, leading to a different surgical approach. Circumferential arterial encasement should be distinguished by the so-called arterial “engulfment”, where the vessel is compressed and dislocated, but the probability of the infiltration of the adventitia is very low [158]. When the lumen of the vessel is not narrowed, the contrast between the enhancing tumor and the blood vessel makes the MRI spin-echo sequences even superior to that of the catheter-based angiography in assessing the arterial encasement of suprasellar and parasellar masses [159] (cases #1, #3–#8). Regardless of the amount of encircling of the parent vessel, the encasement of the perforating arteries is a key concern since the extensive manipulation and dissection may cause intraoperative vessel rupture that is difficult to control or repair through the trans-sphenoidal approach. Furthermore, although vasospasm is reported in open skull base surgery [160], it is also widely documented in the literature with trans-sphenoidal approaches [161,162,163,164]. Hence, the arterial encasement is a call for a transcranial approach.

### 3.6. Brain Parenchymal Invasion

Although the “invasive” pattern of growth of pituitary adenomas is not considered to be related to a malignant behavior [152,165], this finding on preoperative MRI represents an additional difficulty for gross total resection with the trans-sphenoidal corridor [121,128]. It is also confirmed by personal experience (see cases #2–#4). An incomplete resection of the intracranial part of the tumor is known to be associated with a higher incidence of postoperative bleeding. Scheithauer and co-workers found that the frequency of a histologically proven gross invasion of neighboring neurovascular structures, including the brain parenchyma, is more significant in macroadenomas and ACTH, prolactin, and GH-secreting tumors [152]. With particular reference to the dura, Selman et al. reported that the histological evidence of a microscopic invasion is 85% in pituitary adenomas, thus making it more significant than commonly believed [166]. Leptomeningeal infiltration is encountered more frequently than brain parenchymal invasion [152]. The brain parenchymal invasion is assessed by conventional T2-weighted and FLAIR imaging techniques, especially on 3T MRI, and preoperatively depicts the edema caused by leptomeningeal–parenchymal involvement. These data should always be part of the preoperative workup of giant pituitary adenomas since the occurrence of brain invasion may lead to a shift in the choice of the surgical approach toward an open craniotomy. 

### 3.7. Coexisting Cerebral Aneurysms

The reported incidence of intracranial aneurysm associated with a pituitary adenoma ranges between 5.4% and 7.4% [167,168]. This incidence is seven-fold higher for pituitary adenomas than other intracranial tumors [168]. In total, 97% of aneurysms involve the anterior circulation, which adds difficulty to the patient’s management, and in 12% of cases, there are multiple recurrences [167]. GH-secreting tumors have a higher risk of associated intracranial aneurysms because of presumed underlying biological factors [169]. The presence of aneurysms represents an additional risk for pituitary adenoma management, owing to four factors: first, the aneurysm lies in the area surrounding the surgical field; second, the possible variation in blood pressure during general anesthesia may induce aneurysm rupture; third, the risk of radiation-induced intracranial aneurysms after adjuvant radiotherapy for remnants or recurrences is known [170,171]; fourth, de novo aneurysms following gamma knife surgery have been reported, although in the absence of definitive data on the incidence [172]. Regarding general anesthesia, no specific regimens have proven to be superior in protecting against rupture [173]. The literature has reported cases of aneurysm rupture during trans-sphenoidal surgery [174,175,176,177,178]. 

Thus, the need to include vascular studies such as MR angiography. Our group utilizes a dynamic CT angiography [179,180] as part of the preoperative work-up of pituitary tumors requiring surgical resection. In these relatively rare cases, both lesions might be approached transcranially, as suggested [66]. 

### 3.8. Residual and Recurrent Tumors

Recurrent or residual pituitary adenomas previously treated with trans-sphenoidal surgery, open craniotomy, and radiotherapy pose particular challenges due to arachnoid adherences from the previous surgery, increasing the risk of complications. The reported gross total resection rate associated with redo surgery for craniopharyngiomas is 21%, with a mortality rate increasing up to 41% [181,182,183,184]; these data are also theoretically extendible to pituitary adenomas. Gamma knife radiosurgery has been increasingly utilized to manage residual and recurrent tumors. Nevertheless, some essential aspects should be considered before deciding between redo surgery and radiotherapy, particularly in young patients. While considering the current role of radiosurgery in allowing the control of remnants, concerns still exist about its long-term safety. al-Mefty and colleagues reported an overall rate of delayed side effects of 38% after postoperative radiotherapy for benign brain tumors, and 79.3% of the series involved pituitary adenomas [185,186]. These radiation-induced disorders affected a new onset of visual deterioration, cerebral radionecrosis, generalized seizures, cognitive impairments, dementia, pituitary dysfunction, panhypopituitarism, diabetes insipidus, motor deficits, and MRI evidence on parenchymal changes in the brain. Visual complications following radiotherapy for pituitary adenomas are historically considered a source of serious concern [187,188,189,190].

Similarly, hypothalamic and pituitary deficits have been reported [191,192,193,194,195,196,197,198]. The significantly higher risk of radiation-induced second tumors in patients who underwent radiotherapy during adulthood for benign brain tumors constitutes further alarm [185]. They were reported to be malignant in 86% of cases, including high-grade meningiomas, sarcomas (particularly fibrosarcomas), and high-grade gliomas [185]. The latency for sarcomas is as short as one year [199]. Two autoptic cases of radiation-induced primitive neuroectodermal tumors (PNETs) have been reported [200]. The risk of a sarcomatous transformation of pituitary adenoma following radiotherapy is also well known, although underrated [201,202,203,204,205,206,207,208,209,210,211,212]. For these reasons, several authors do not recommend routine postoperative radiotherapy for remnants of incomplete resected pituitary adenomas [213,214]. We embrace the same policy, and the management of reported cases #1 and #7 provides examples of how a staged trans-sphenoidal–transcranial microsurgical strategy allows for a near or gross total resection with good patient outcomes. Risks of radiation-induced changes in the cerebral arterial wall, including intracranial aneurysms, vessel wall thickening, thrombosis, luminal occlusion, and telangiectases, exist [170,171,215]. A further critical point to consider regards the need to pursue an endocrinological cure for the illness through a radical resection in de novo and recurrent nonfunctioning pituitary adenomas. Radiosurgery should be considered for occasional recurrences where the patient is medically unfit for surgical interventions, especially in older patients with multiple comorbidities. 

Nicholas et al. reported that close observation and postponement of radiotherapy until the evidence of progression is a good and safe option for initial adenoma sizes ranging between 2 cm and 4 cm because most of these patients do not require further treatment [216]. As a rule, a primary surgical route is associated with a safer and higher likelihood of gross total resection. In cases of residual or recurrent tumors previously treated with trans-sphenoidal surgery or radiotherapy, open transcranial routes allow for multiple working corridors, safer sharp dissection, and better hemostasis [124]. The selection of an operative approach depends upon extension into the CS and the suprasellar area. In contrast, the sidedness of the approach in transcranial approaches performed after incomplete trans-sphenoidal resection is chosen according to the direction of the tumor extension. Some propose approaching the tumor from the side of the optic nerve dysfunction. The right side is preferred for a right-handed surgeon facing a midline tumor. The advantages of extended subfrontal and COZ approaches have been advocated for tumors involving the medial and lateral walls of the CS, respectively [141,142,149,217,218,219]. Simultaneous trans-sphenoidal–transcranial approaches for resectioning large de novo or recurrent tumors are non-preferred because of the reported high success rate of the staged trans-sphenoidal–transcranial approach [220]. 

The aggressive recurrence of pituitary adenomas, particularly ACTH-secreting tumors, could extend and grow in a cranial location that is only reachable by the transcranial approach. In these cases, it is clear that the trans-sphenoidal approach is not applicable.

#### Case #8: Aggressive ACTH Tumor with Multiple Recurrences

An unfortunate patient had an aggressive ACTH tumor with several recurrences despite repeated surgery, radiation, and medical treatment. The location of her recurrences dictated several different cranial approaches, including the petrosal approach on the left side, the anterior petrosal on the right side, and suboccipital craniotomies (Figure 26).

### 3.9. Coexisting Isolated Pathologies of The Sphenoid Sinus

Concurrent pathologies of the sphenoid sinus limit the practicality of the trans-sphenoidal corridor to intracranial pathologies, often leading to open craniotomy being favoured. The spectrum of these lesions mainly involves inflammatory or neoplastic diseases. Bacterial and fungal sinusitis have average incidence rates of 38% and 20%, respectively [221,222,223]. Neoplasms account for 16%, whereas mucoceles, cysts, and sphenochoanal polyps are rare [221,222,223].

### 3.10. Postoperative Pituitary Apoplexy after Trans-Sphenoidal Surgery

Post-operative pituitary apoplexy is considered one of the most severe complications of giant adenoma surgery. Its estimated incidence is 5.6%, whereas the overall mortality is 42%. A subtotal resection of the tumor (<90%) is associated with this occurrence [224,225]. When apoplexy is secondary to trans-sphenoidal surgery, a transcranial approach allows for a total hematoma evacuation and better hemostasis. Indications for surgery should be based on the evidence of a deteriorating visual function for small-volume bleedings and classic algorithms for managing spontaneous intracranial hemorrhages in the case of a significant mass effect [226,227].

## 4. Conclusions

Despite the popularity and outcomes achieved over the last two decades by endonasal trans-sphenoidal endoscopic techniques, transcranial approaches continue to hold a critical role in managing giant and more complex pituitary adenomas characterized by a wide intracranial extension, allowing gross total resection with good outcomes.

## 5. Future Directions

As dopamine agonist treatment has offered a breakthrough in the treatment of prolactinomas, fostering the hope and anticipation that similar medical treatments will be as effective in the treatment of other hypersecreting adenomas which will mark another triumph in the management of pituitary adenomas and make surgery unneeded but for particular circumstances. Future directions of giant pituitary adenoma surgery involve improvements in preoperative evaluation, planning, intraoperative courses, and postoperative management. The advancement of neuroimaging techniques will allow us to precisely estimate the consistency of the tumor and the invasion of the CS, dura, leptomeningeal, and brain parenchyma [228]. The better spatial resolution of the tumor–neurovascular structure interface in case of their encasement will allow the grade of invasiveness of the adenoma to be anticipated towards the adventitia and perineurium. Radiomics is increasing its sensitivity and specificity [228,229,230]. This information is critical in choosing and tailoring the most suitable surgical approach. The routine use of high-field and ultra-high-field intraoperative magnetic resonance imaging (iMRI) has been proposed to increase the EOR rate, while at the same time decreasing the recurrence rate, meaning that it will probably become a mainstay in the future [231]. Similarly to gliomas, the use of fluorescent tracers, such as 5-aminolevulinic acid (5-ALA), indocyanine green (ICG), and OTL38, has been proposed to maximize the resection [232,233]. The first prototyped robots for endonasal trans-sphenoidal surgery have come to appear with theoretically interesting future applications, but need validation on a large scale [234,235,236,237]. Molecular biology will play an increasingly critical part in the early identification of more aggressive pituitary adenomas from a genetic point of view. The exact molecular biology will also definitively clarify the long-term effects of radiotherapy on residual tumors, allowing for a better definition of its safety profile.

## Figures and Tables

**Figure 1 cancers-15-02235-f001:**
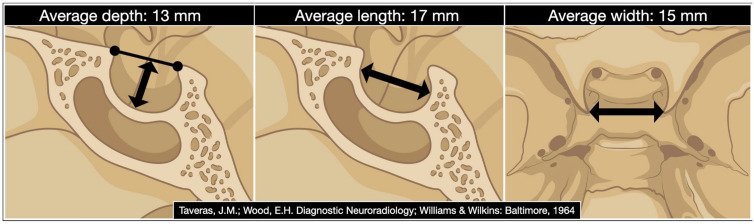
The digital drawing shows the sellar region’s average depth, length, and width. (Date derived from Taveras JM, Wood EH. Diagnostic Neuroradiology. Baltimore: Williams and Wilkins; 1964 [73]).

**Figure 2 cancers-15-02235-f002:**
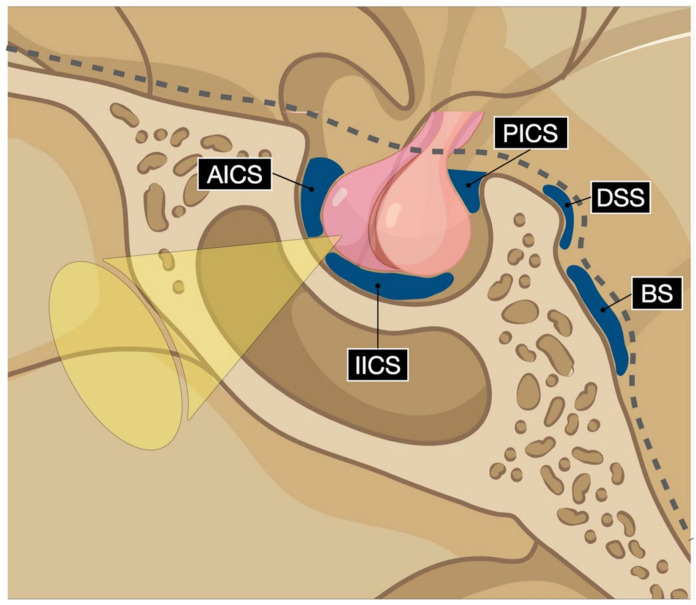
Digital drawing schematizes the intercavernous sinuses’ possible vascular pattern. AICS: anterior intercavernous sinus; IICS: inferior intercavernous sinus; PICS: posterior intercavernous sinus; DSS: dorsal sphenoid sinus; BS: basilar sinus.

**Figure 3 cancers-15-02235-f003:**
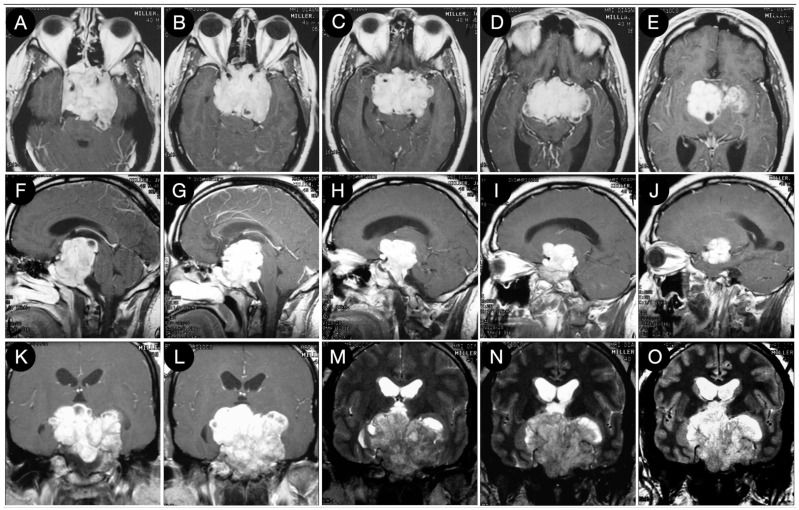
Preoperative axial (**A**–**E**), sagittal (**F**–**J**), and coronal (**K**–**O**) T1-weighted contrast-enhanced MRI. Scale bar: 5 cm.

**Figure 4 cancers-15-02235-f004:**
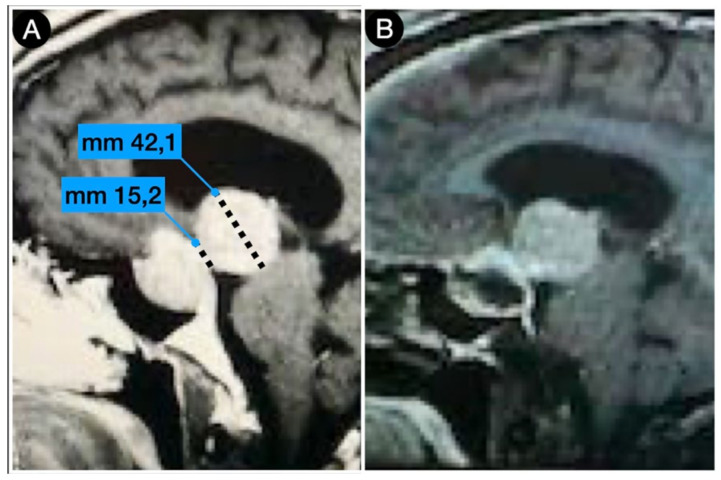
Preoperative (**A**) and postoperative (**B**) (post-trans-sphenoidal approach) sagittal T1-weighted contrast-enhanced MRI.

**Figure 5 cancers-15-02235-f005:**
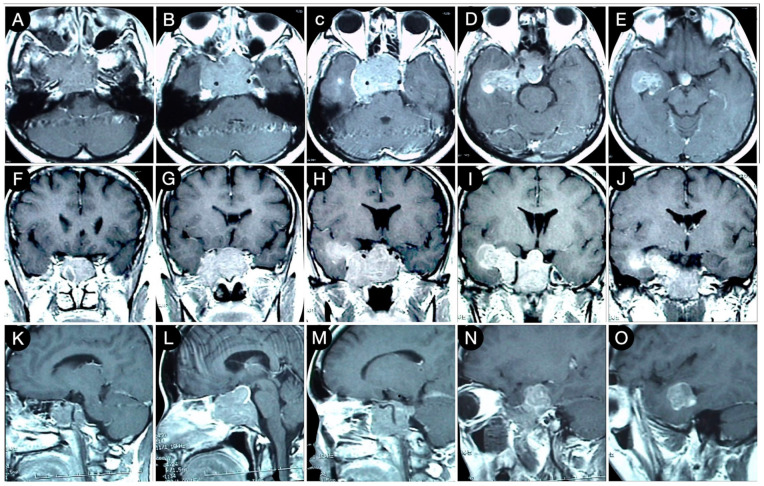
Preoperative axial (**A**–**E**), coronal (**F**–**J**), and sagittal (**K**–**O**) T1-weighted contrast-enhanced MRI. Scale bar: 5 cm.

**Figure 6 cancers-15-02235-f006:**
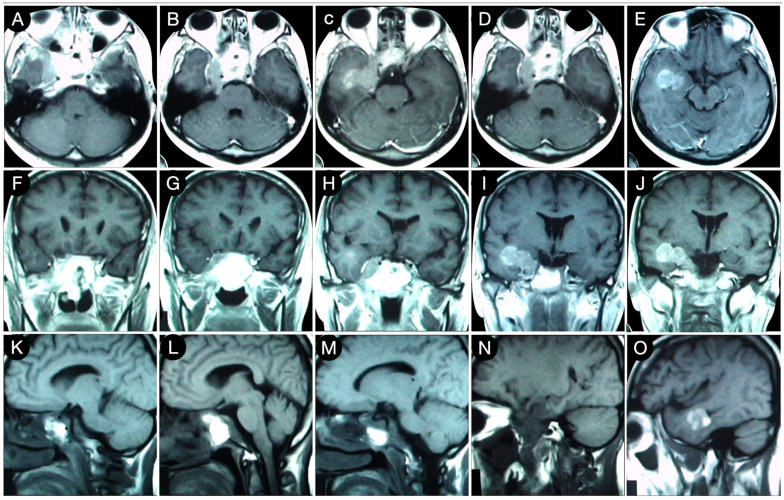
Axial (**A**–**E**), coronal (**F**–**J**), and sagittal (**K**–**O**) T1-weighted contrast-enhanced MRI after the trans-sphenoidal resection of the sellar part of the tumor. Scale bar: 5 cm.

**Figure 7 cancers-15-02235-f007:**
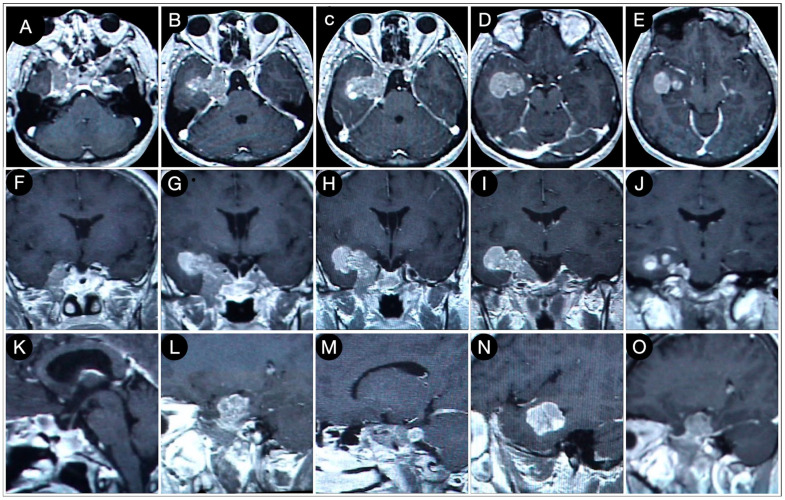
Axial (**A**–**E**), coronal (**F**–**J**), and sagittal (**K**–**O**) T1-weighted contrast-enhanced MRI performed six months after the trans-sphenoidal resection of the sellar part of the tumor. Scale bar: 5 cm.

**Figure 8 cancers-15-02235-f008:**
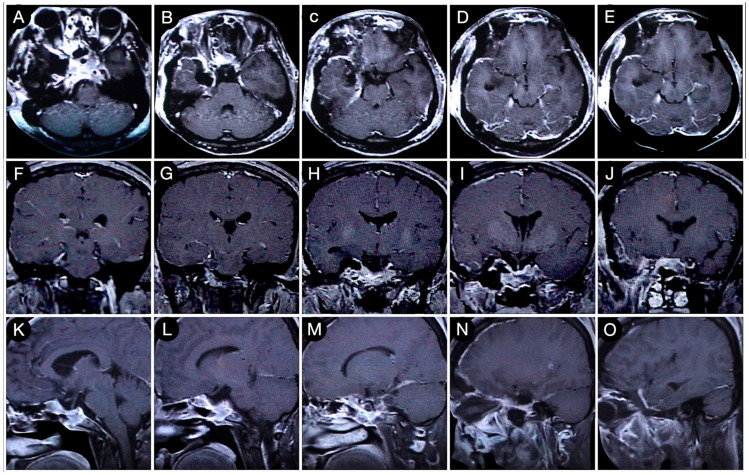
Axial (**A**–**E**), coronal (**F**–**J**), and sagittal (**K**–**O**) T1-weighted contrast-enhanced MRI after the COZ approach for the resection of the intracranial part of the tumor. Scale bar: 5 cm.

**Figure 9 cancers-15-02235-f009:**
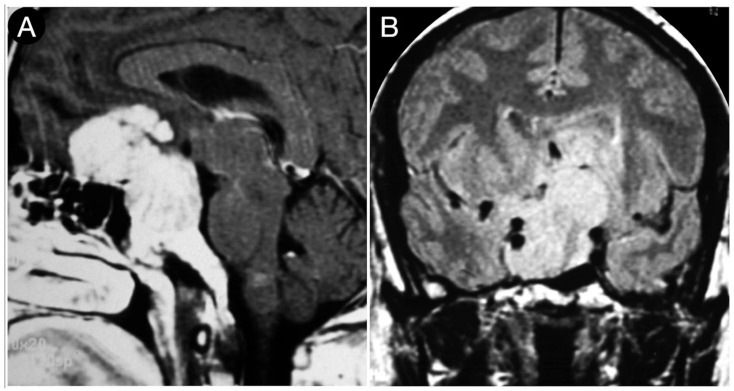
Preoperative sagittal (**A**) and coronal (**B**) T1-weighted contrast-enhanced MRI. Scale bar: 5 cm.

**Figure 10 cancers-15-02235-f010:**
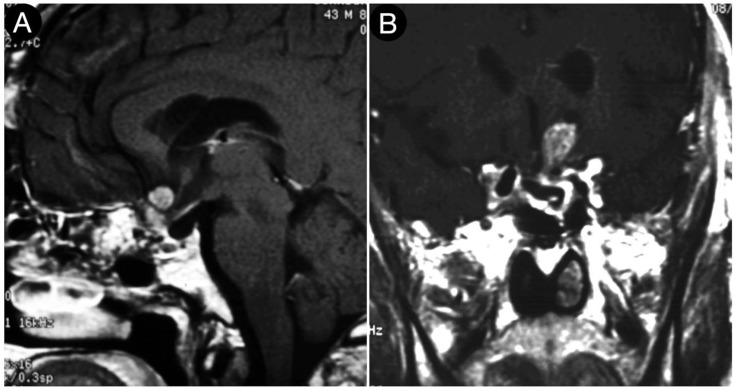
Postoperative sagittal (**A**) and coronal (**B**) T1-weighted contrast-enhanced MRI. Scale bar: 5 cm.

**Figure 11 cancers-15-02235-f011:**
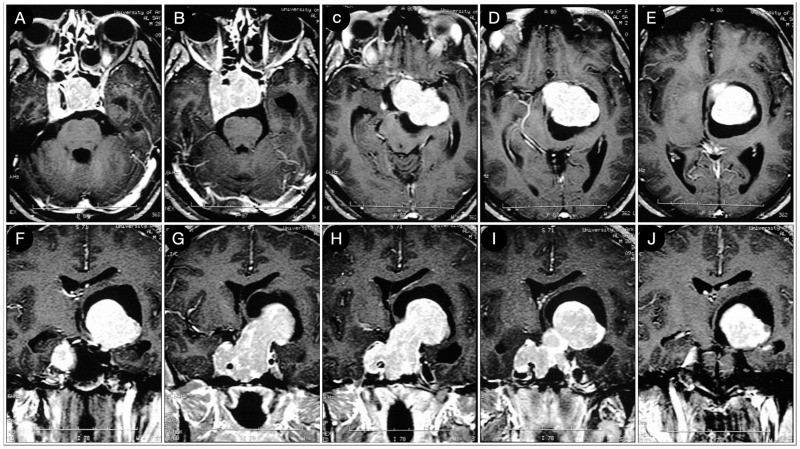
Preoperative axial (**A**–**E**) and coronal (**F**–**J**) T1-weighted contrast-enhanced MRI. Scale bar: 5 cm.

**Figure 12 cancers-15-02235-f012:**
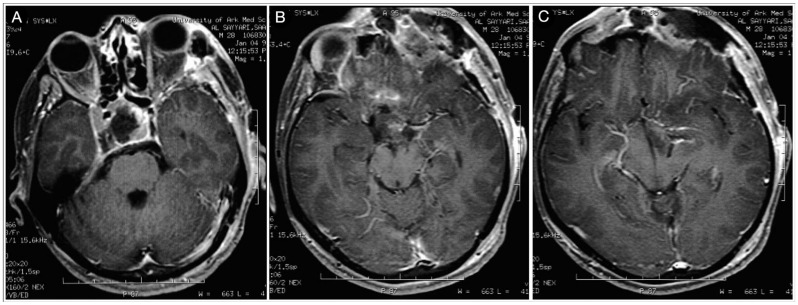
(**A**–**C**) Postoperative axial T1-weighted contrast-enhanced MRI. Scale bar: 5 cm.

**Figure 13 cancers-15-02235-f013:**
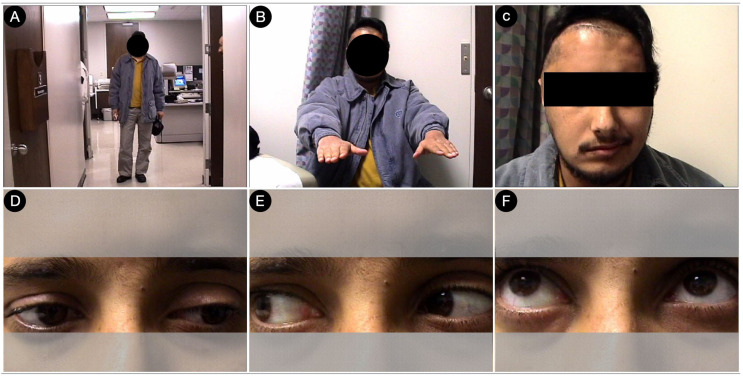
Postoperative pictures of the patients showing the motor (**A**,**B**), facial nerve (**C**), and oculomotor (**D**–**F**) functions.

**Figure 14 cancers-15-02235-f014:**
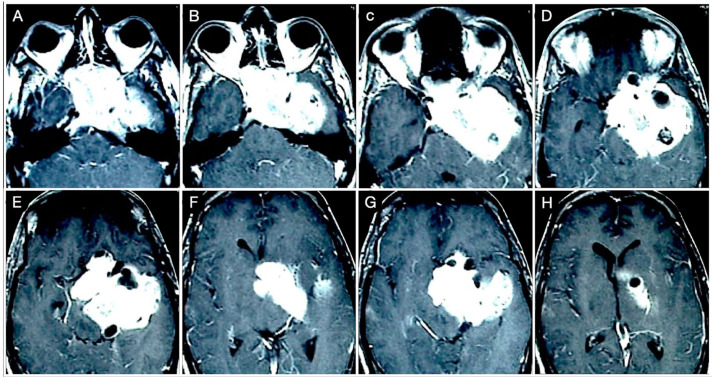
Preoperative axial (**A**–**H**) T1-weighted contrast-enhanced MRI. Scale bar: 5 cm.

**Figure 15 cancers-15-02235-f015:**
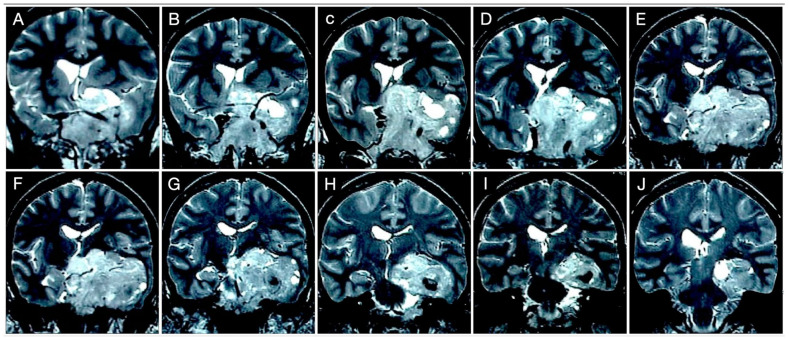
Preoperative coronal (**A**–**J**) T2-weighted contrast-enhanced MRI. Scale bar: 5 cm.

**Figure 16 cancers-15-02235-f016:**
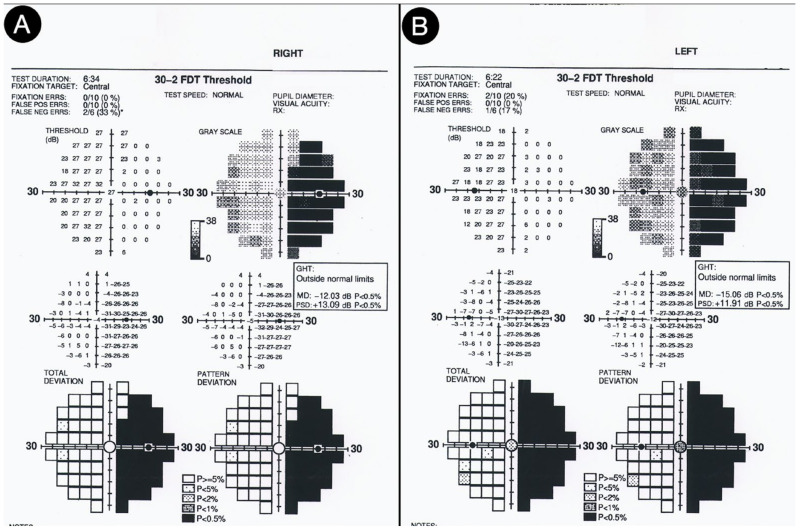
Preoperative visual field test of the right (**A**) and left (**B**) eye.

**Figure 17 cancers-15-02235-f017:**
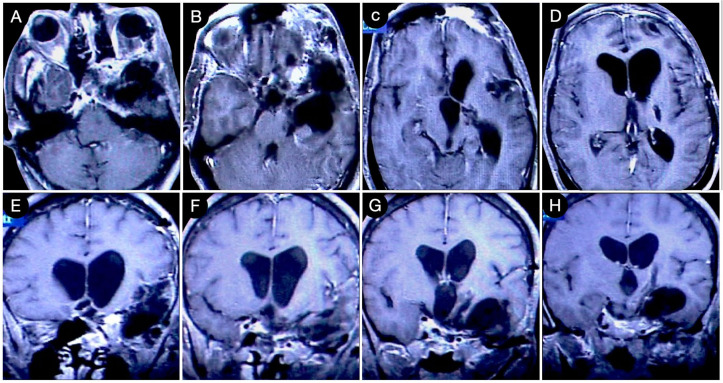
Postoperative axial (**A**–**D**) and coronal (**E**–**H**) T1-weighted contrast-enhanced MRI. Scale bar: 5 cm.

**Figure 18 cancers-15-02235-f018:**
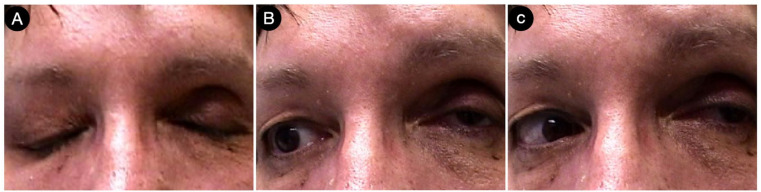
(**A**–**C**) Postoperative pictures of the patients, showing the oculomotor function.

**Figure 19 cancers-15-02235-f019:**
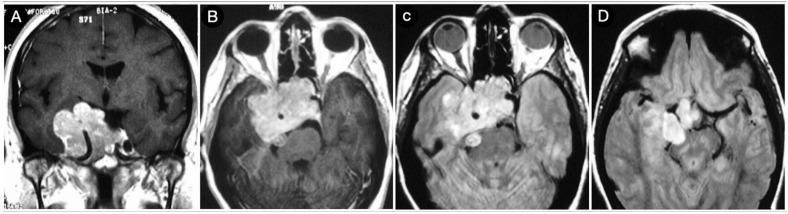
Preoperative coronal (**A**) and axial (**B**–**D**) T1-weighted contrast-enhanced MRI. Scale bar: 5 cm.

**Figure 20 cancers-15-02235-f020:**
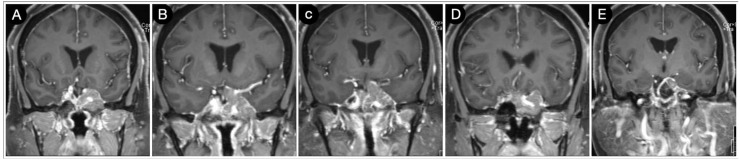
(**A**–**E**) Preoperative coronal T1-weighted contrast-enhanced MRI. Scale bar: 5 cm.

**Figure 21 cancers-15-02235-f021:**
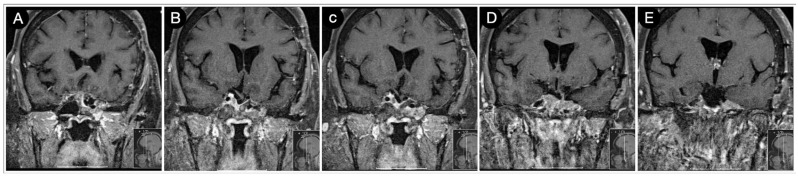
(**A**–**E**) Postoperative coronal T1-weighted contrast-enhanced MRI. Scale bar: 5 cm.

**Figure 22 cancers-15-02235-f022:**
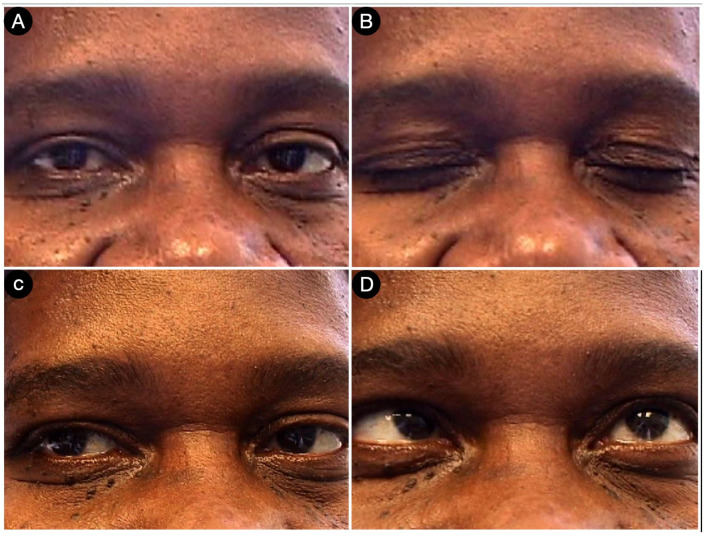
(**A**–**D**) Postoperative pictures of the patients, showing the abducens nerve palsy.

**Figure 23 cancers-15-02235-f023:**
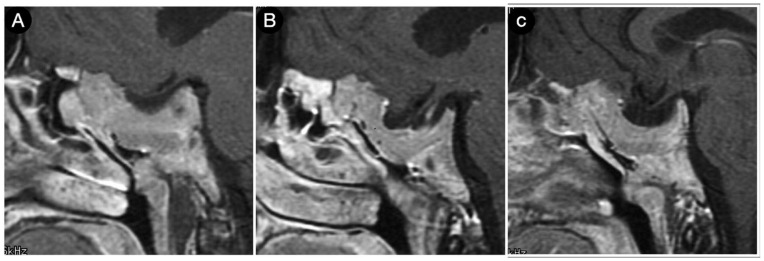
(**A**–**C**) sagittal T1-weighted contrast-enhanced MRI of the sellar region. Scale bar: 5 cm.

**Figure 24 cancers-15-02235-f024:**
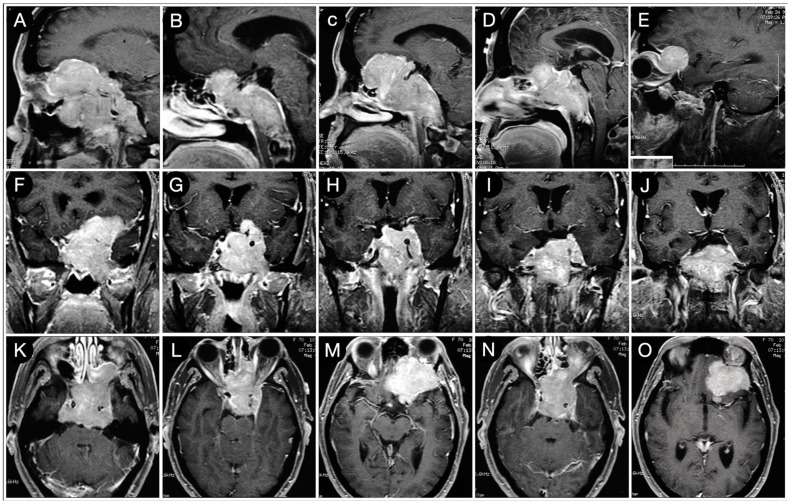
Preoperative sagittal (**A**–**E**), coronal (**F**–**J**), and axial (**K**–**O**) T1-weighted contrast-enhanced MRI. Scale bar: 5 cm.

**Figure 25 cancers-15-02235-f025:**
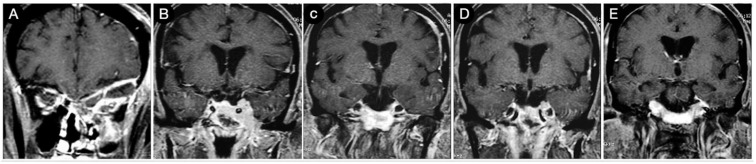
(**A**–**E**) Postoperative T1-weighted contrast-enhanced MRI. Scale bar: 5 cm.

**Figure 26 cancers-15-02235-f026:**
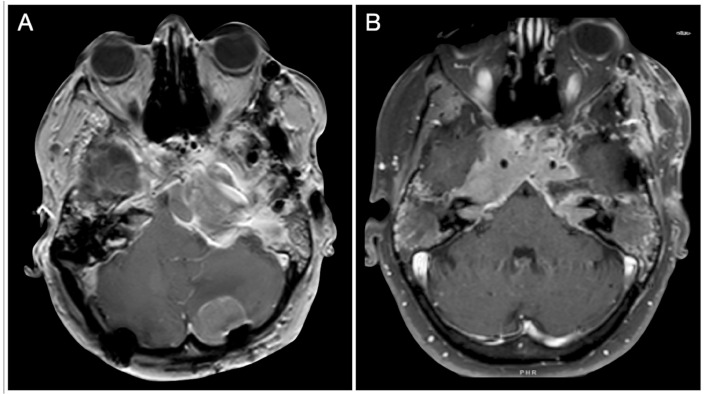
(**A**,**B**) Axial T1-weighted contrast-enhanced MRI. Scale bar: 5 cm.

## Data Availability

All data are included in the main text.

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
