# Peer review of "The Party Wall: Redefining the Indications of Transcranial Approaches for Giant Pituitary Adenomas in Endoscopic Era"

_cancers, 2023, doi:10.3390/cancers15082235_

Round 1
Reviewer 1 Report
The authors present a narrative Review on “the Indications of Transcranial Approaches for Giant Pituitary Adenomas in Endoscopic” citing 322 references. In the Abstract they state that “A critical appraisal of the personal series of the senior author (O.A.) was performed to characterize the patient factors and the tumor’s pathological anatomy features that endorse a cranial approach”. The Review is separated in the sections Introduction, Anatomical factors (1 case), Tumor-related factors (7 cases), Conclusions, Future directions.
In the area of systematic reviews, narrative reviews need substantial justification. The most relevant point is the extensive experience of the senior author in skull base surgery. To pass his comprehensive knowledge to future generations substantiates this manuscript. Nevertheless, in order to improve comprehensibility, I would suggest re-structuring the manuscript.
(1) The assignment of cases to either Anatomical- of tumor related facts is hard to understand and should be clarified.
(2) Why did the authors choose the respective 8 cases? I would recommend putting these cases in the context of the total series of the senior author respectively cases published in the past.
(3) I would recommend reducing the 322 references to the essential ones.
Minor comments
· Simple Summary duplicates Abstract.
· L62/73 Introduction of abbreviation Tsph/cra: Really necessary? To me, it interfers with comprehensibility.
· L94 Using “party wall” with two different meanings is confusing, i.e. “giant pituitary adenomas represent a party wall between Tsph and Tcra approaches”.
· L95 “The explosive evolution of endoscopic transnasal Tsph surgery and its impressive results in pituitary tumors have overshadowed” I would prefer a little less dramatic lyric language in order to transfer scientific content.
Author Response
Ref.: Ms. No. Cancers-2262575
The Party Wall: Redefining the Indications of Transcranial Approaches for Giant Pituitary Adenomas in Endoscopic Era
Response to Reviewer
We want to thank the kind Reviewer for his comments and suggestions, which have been precious in improving our manuscript's quality and clarity.
We have made the suggested revisions to our manuscript and reported an itemized, point-by-point response to the Reviewer's kind remarks.
All the manuscript modifications have been highlighted in the manuscript file with the Track-Changes Word function.
- The authors present a narrative Review on "the Indications of Transcranial Approaches for Giant Pituitary Adenomas in Endoscopic" citing 322 references. In the Abstract they state that "A critical appraisal of the personal series of the senior author (O.A.) was performed to characterize the patient factors and the tumor's pathological anatomy features that endorse a cranial approach". The Review is separated in the sections Introduction, Anatomical factors (1 case), Tumor-related factors (7 cases), Conclusions, Future directions.
In the area of systematic reviews, narrative reviews need substantial justification. The most relevant point is the extensive experience of the senior author in skull base surgery. To pass his comprehensive knowledge to future generations substantiates this manuscript. Nevertheless, in order to improve comprehensibility, I would suggest re-structuring the manuscript.
We want to thank the kind Reviewer for the comment. Passing the knowledge from the senior author's (OA) long-lasting experience to future generations is exactly the aim of the article.
- The assignment of cases to either Anatomical- of tumor related facts is hard to understand and should be clarified.
Thank you for this precious suggestion. We totally agree with the misleading effect of the used terms. In the revised version of the manuscript, we have changed the subtitle "Anatomical factors" to "Factors Related to The Inter-individual Anatomical Variability of the Sellar and Parasellar Area." The subtitle "tumor-related factors" has also been changed to "Factors Related to The Tumor Features."
- Why did the authors choose the respective 8 cases? I would recommend putting these cases in the context of the total series of the senior author respectively cases published in the past.
We agree with the Reviewer. Nevertheless, we decided to select and present multiple illustrative cases as each of them highlights a specific aspect of surgical management of giant pituitary adenomas. A further reason is that all the reported cases haven't been published before. Therefore, it was necessary to put them in the present narrative review.
- I would recommend reducing the 322 references to the essential ones.
We also totally agree on the need to reduce the number of references. Accordingly, in the revised manuscript, the number of references has been decreased to 238 versus 322 in the previous version.
Minor comments
- Simple Summary duplicates Abstract.
The Simple Summary, apart from the Abstract, is specifically required by the Journal Authors' guidelines.
- L62/73 Introduction of abbreviation Tsph/cra: Really necessary? To me, it interfers with comprehensibility.
Thank you also for this point. In the Revised version, we have avoided the abbreviations Tsph and Tcra.
- L94 Using "party wall" with two different meanings is confusing, i.e. "giant pituitary adenomas represent a party wall between Tsph and Tcra approaches".
We changed the term "party wall" in L94 to avoid sources of confusion.
- L95 "The explosive evolution of endoscopic transnasal Tsph surgery and its impressive results in pituitary tumors have overshadowed" I would prefer a little less dramatic lyric language in order to transfer scientific content.
Based on the kind Reviewer's suggestion, we have eliminated adjectives such as "explosive," "impressive," etc.
We hope that the revisions to our manuscript may be adequate and satisfying as all the valuable suggestions of the kind Reviewer have been respected.
We want to thank the kind Reviewer and the Editor again for the valuable suggestions, which have been paramount for improving the overall clarity and quality of the article.
The Authors
Reviewer 2 Report
The study is very interesting. It clearly and exhaustively explains the indications and the validity of the transcranial approach in the treatment of giant pititary adenomas.
Additional comments:
In this narrative review, the authors report the indications and validity of the transcranial approach in the treatment of giant pituitary adenomas. The study appears to be complete, exhaustive and with interesting data on the subject. The cases described confirm the validity of the transcranial approach in selected cases. The conclusions are pertinent. The references, even if numerous, appear appropriate. The figures are understandable and useful for understanding the text.
Author Response
Ref.: Ms. No. Cancers-2262575
The Party Wall: Redefining the Indications of Transcranial Approaches for Giant Pituitary Adenomas in Endoscopic Era
Response to Reviewer
We want to thank the kind Reviewer for his comments and suggestions, which have been precious in improving our manuscript's quality and clarity.
All the manuscript modifications have been highlighted in the manuscript file with the Track-Changes Word function.
- The study is very interesting. It clearly and exhaustively explains the indications and the validity of the transcranial approach in the treatment of giant pituitary adenomas.
Additional comments:
- In this narrative review, the authors report the indications and validity of the transcranial approach in the treatment of giant pituitary adenomas. The study appears to be complete, exhaustive and with interesting data on the subject. The cases described confirm the validity of the transcranial approach in selected cases. The conclusions are pertinent. The references, even if numerous, appear appropriate. The figures are understandable and useful for understanding the text.
We want to thank the kind Reviewer for this comment, which made us proud. Passing the knowledge from the senior author's (OA) long-lasting experience to future generations is exactly the aim of the article.
We totally agree on the need to reduce the number of references. Accordingly, in the revised manuscript, the number of references has been decreased to 238 versus 322 in the previous version.
We want to thank the kind Reviewer and the Editor again for their comments and valuable suggestions.
The Authors